# Therapeutic Effects of Aβ-Specific Regulatory T Cells in Alzheimer’s Disease: A Study in 5xFAD Mice

**DOI:** 10.3390/ijms25020783

**Published:** 2024-01-08

**Authors:** Seon-Young Park, Juwon Yang, Hyejin Yang, Inhee Cho, Jae Yoon Kim, Hyunsu Bae

**Affiliations:** 1Department of Science in Korean Medicine, College of Korean Medicine, Graduate School, Kyung Hee University, 26 Kyungheedae-ro, Dongdaemun-gu, Seoul 02447, Republic of Korea; psys12@naver.com (S.-Y.P.); emilly86@naver.com (H.Y.); 2Department of Korean Medicine, College of Korean Medicine, Graduate School, Kyung Hee University, 26 Kyungheedae-ro, Dongdaemun-gu, Seoul 02447, Republic of Korea; jwyang@khu.ac.kr (J.Y.); okoriental@naver.com (I.C.); 3Institute of Life Science & Biotechnology, VT Bio. Co., Ltd., 16 Samseong-ro 76-gil, Gangnam-gu, Seoul 06185, Republic of Korea; jykim@vtbio.co.kr

**Keywords:** Alzheimer’s disease, regulatory T cells, Amyloid-β

## Abstract

The aging global population is placing an increasing burden on healthcare systems, and the social impact of Alzheimer’s disease (AD) is on the rise. However, the availability of safe and effective treatments for AD remains limited. Adoptive Treg therapy has been explored for treating neurodegenerative diseases, including AD. To facilitate the clinical application of Treg therapy, we developed a Treg preparation protocol and highlighted the therapeutic effects of Tregs in 5xFAD mice. CD4^+^CD25^+^ Tregs, isolated after Aβ stimulation and expanded using a G-rex plate with a gas-permeable membrane, were adoptively transferred into 5xFAD mice. Behavioral analysis was conducted using Y-maze and passive avoidance tests. Additionally, we measured levels of Aβ, phosphorylated tau (pTAU), and nitric oxide synthase 2 (NOS2) in the hippocampus. Real-time RT-PCR was employed to assess the mRNA levels of pro- and anti-inflammatory markers. Our findings indicate that Aβ-specific Tregs not only improved cognitive function but also reduced Aβ and pTAU accumulation in the hippocampus of 5xFAD mice. They also inhibited microglial neuroinflammation. These effects were observed at doses as low as 1.5 × 10^3^ cells/head. Collectively, our results demonstrate that Aβ-specific Tregs can mitigate AD pathology in 5xFAD mice.

## 1. Introduction

Presently, over 55 million individuals worldwide are affected by dementia, with nearly 10 million new cases emerging annually. Alzheimer’s disease (AD) stands as the predominant form, constituting 60–70% of all dementia cases [1].

The neuropathological characteristics of AD encompass chronic neuroinflammation linked to the extracellular deposition of amyloid-β (Aβ), intraneural neurofibrillary tangles, astrocytosis, and microgliosis [2,3]. In central nervous system (CNS) disorders, including AD, neuroinflammation plays a role in the progression of pathology by influencing blood-brain barrier (BBB) integrity [4,5]. Amyloid-β and phosphorylated tau, distinctive AD proteins, form extracellular neuritis plaques and intracellular neurofibrillary tangles, representing focal points in AD research [6].

Regulatory T cells (Tregs) are indispensable for suppressing moderate immune responses and maintaining immune homeostasis. Several mechanisms of Treg-mediated suppression include the secretion of immunosuppressive cytokines by the Treg, cell-contact-dependent suppression, and functional modification or killing of antigen-presenting cells [7]. Their significance extends to neuroinflammation in the CNS [8]. Notably, impaired Treg function has been identified in patients with multiple sclerosis (MS), and Treg accumulation has been linked to recovery in experimental autoimmune encephalomyelitis (EAE) models [9,10]. Similarly, a reduction in the number of Tregs has been reported in patients with mild AD [11].

Treg-based therapies have emerged as promising strategies, broadly categorized into two approaches. One involves the administration of immunomodulatory interventions designed to foster the expansion and enhance the function of Tregs in vivo. The alternative approach entails the adoptive transfer of Tregs that have been expanded in vitro. Tregs themselves can be further classified into antigen-specific and expanded polyclonal Tregs [12]. Polyclonal Treg-based cell therapy has successfully transitioned into clinical practice, with trials demonstrating the safety and efficacy of this Treg therapeutic approach. Moreover, in animal models, antigen-specific Tregs have shown functional superiority compared to polyclonal Tregs [13].

In our previous study, we manufactured antigen-specific Tregs and demonstrated their effectiveness in treating CNS disorders in a mouse model. We observed that the adoption of Aβ-specific Tregs attenuates the progression of AD [14], and administration of alpha-synuclein-specific Tregs ameliorates PD progression [15].

5xFAD and 3xTg represent the most commonly employed mouse models in AD research. 5xFAD mice bear mutations in the APP and presenilin (PS) genes, exhibiting amyloid deposition as early as 2 months. They swiftly manifest key AD pathology features, including neuronal loss and memory impairment, in contrast to 3xTg mice. Cognitive impairment resulting from these alterations can be evaluated through behavioral assessments, such as maze tests [16,17,18]. In our previous study, we manufactured antigen-specific Tregs and demonstrated their effectiveness in treating CNS disorders.

In a previous investigation, we used the 3xTg AD mouse model to explore the neuroprotective effect of improving the cognitive function of Tregs in AD [19]. Based on the results, we employed young 5xFAD mice and successfully replicated the therapeutic effects of Tregs.

In the present study, we employed a young AD mouse model to examine the impact of Aβ-specific Tregs and refined the Treg expansion protocol using Gas-permeable Rapid Expansion (G-Rex). Our objective was to enhance the reproducibility of experiments involving human Tregs and propel them toward becoming a viable treatment option for AD.

## 2. Results

### 2.1. Ex Vivo Expansion of Aβ-Specific Treg

For the development of a reproducible method for expanding Aβ-specific Treg for clinical applications, CD4^+^CD25^+^ Tregs were initially isolated and subsequently expanded using G-rex plates in vitro. We obtained splenocytes from wild-type (WT) mice and depleted CD8 cells to obtain CD8-cells. For antigen presentation, CD8-cells were incubated with Aβ and bee venom phospholipase A2 for a duration of 4 days and CD4^+^CD25^+^ Treg was isolated and expanded for 2 weeks (details provided in ‘Section 4’) (Figure 1A).

The purity of the isolated cells was assessed by flow cytometry. CD8-cells were stained with FITC-CD8 after CD8 depletion and, as a result, it was confirmed that most CD8+ cells were depleted. After Treg isolation, the isolated Tregs were stained with PE-Cy5-CD4 and APC-cy7-CD25, and the purity of Tregs was confirmed to be over 97%. Tregs cultured in G-rex were confirmed to be expanded by staining with PE-Cy5-CD4 and APC-cy7-CD25 (Figure 1B).

### 2.2. Treg Transfer Ameliorates Cognitive Dysfunction in 5xFAD Mice

To evaluate the therapeutic effects of ex vivo-expanded Tregs in AD, groups of 5xFAD mice were administered 1.5 × 10^3^, 10^4^, and 10^5^ ex vivo-expanded Tregs. Behavioral tests were conducted after eight weeks (Figure 2A). Y-maze alternation tests were employed to assess cognitive function (Figure 2B). Mice were allowed to explore all three arms of the Y-maze freely, and spontaneous alternation was calculated. The ex vivo-expanded Tregs improved alternation scores, which were reduced in 5xFAD mice. In the passive avoidance experiment, mice were trained to escape the conditioning chamber with the foot shock. The test result showed that the escape latency of passive avoidance in the ex vivo-expanded Tregs injected group was significantly increased compared to the 5xFAD mice (Figure 2C). Notably, these effects on cognitive function were dose-dependent.

### 2.3. Treg Transfer Reduces Accumulation of aβ and Phosphorylated-Tau in 5xFAD Mice

The accumulation of Aβ stands as a prominent pathological hallmark in AD [20]. We extracted the brains of mice that completed behavioral testing for 8 weeks after infusion of Tregs expanded at various doses and quantified Aβ levels in the hippocampal CA1 region using immunofluorescence (Figure 3A). Aβ fluorescence intensity increased in the 5xFAD mice compared to WT (Figure 3B) but significantly decreased in the group transferred ex vivo-expanded Tregs (*p* < 0.001).

Additionally, we assessed the levels of phosphorylated tau (p-tau), another hallmark of AD (Figure 4A). In 5xFAD mice, p-tau deposition was elevated compared to that in WT group, yet this elevation was mitigated by the transfer of ex vivo-expanded Tregs (Figure 4B). The fluorescence intensity of p-tau in groups of 1.5 × 10^4^ and 10^5^ ex vivo-expanded Tregs transfered with 5xFAD mice is similar to WT. These findings imply that ex vivo-expanded Tregs play a role in ameliorating AD pathology.

### 2.4. Treg Transfer Modulates Neuroinflammation of 5xFAD Mice

This study investigates the impact of ex vivo-expanded Tregs on neuroinflammation. We quantified the levels of NOS2 in the hippocampus using immunofluorescence (Figure 5A). In 5xFAD mice, NOS2 deposition was elevated compared to that in WT group, yet the transfer of ex vivo-expanded Tregs mitigated this elevation. Additionally, we assessed the mRNA expression of pro-inflammatory markers (NOS2, TNF-α, IL-1β, and IL-6) and anti-inflammatory markers (Arg1 and Mrc1) by RT-qPCR (Figure 5B). The transfer of ex vivo-expanded Tregs effectively suppresses the pro-inflammatory response while bolstering the anti-inflammatory response. These findings substantiate the hypothesis that ex vivo -expanded Tregs play a role in alleviating neuroinflammation in mice models of AD.

## 3. Discussion

The etiology of AD has been attributed to various pathological processes, with the Aβ toxicity and amyloid cascade hypotheses emerging prominently. These hypotheses propose that the accumulation of Aβ peptide and synaptic alterations in the brain plays a central role in AD pathology [21,22]. Studies have demonstrated that the extracellular deposition of Aβ peptide leads to the formation of senile plaques, while intracellular neurofibrillary tangles, composed of hyperphosphorylated tau protein, also develop [20,23]. Despite extensive research, drug trials targeting these pathological mechanisms have shown limited success.

Aducanumab is a monoclonal antibody designed to target amyloid beta and was approved for medical use by the U.S. Food and Drug Administration (FDA) in June 2021. Although it is the first treatment approved for Alzheimer’s disease, it is controversial due to safety issues such as low clinical efficacy and side effects such as intracerebral microhemorrhage and edema [24]. Lecanemab is also a monoclonal antibody drug targeting amyloid beta and was approved by the FDA in July 2023. This is the second drug approved to reduce amyloid markers and significantly slow cognitive decline in early-stage Alzheimer’s disease. However, as safety concerns have been raised about risks such as cerebral hemorrhage, long-term clinical trials are needed to confirm the efficacy and safety of lecanemab in early Alzheimer’s disease [25,26]. Overall, there is a growing need for effective AD treatments without side effects.

The efficacy of Treg immunotherapy has been demonstrated in various disease models, including graft-versus-host disease, colitis, type 1 diabetes, and multiple sclerosis [27,28,29,30]. Specifically, the expansion of antigen-specific Tregs is a potential strategy currently under exploration in preclinical trials [31]. Advancing the clinical application of Treg therapy is crucial to enhance its efficiency and establish reproducibility in patients. This Treg expansion protocol using G-rex can be used for reproducibility of human Tregs isolated from peripheral blood mononuclear cells (hPBMC). Notably, in this study, it is significant that even a low dose of 1.5 × 10^3^ Aβ-specific Tregs demonstrated neuroprotective effects in 5xFAD mice.

Our research consistently focuses on Treg cell therapy for the treatment of AD. In 2016, we highlighted the potential of Treg therapy in AD [19] and further developed a Treg treatment strategy using Aβ antigen presentation. Our findings demonstrate that the administration of Aβ antigen-specific Tregs modulates the inflammatory state in AD [14]. In this study, we attempted antigen presentation and expansion using G-rex plates after Treg isolation, complementing the Aβ-specific Treg production method in the previous study. This research reinforces the effectiveness of Aβ-specific Treg therapy for AD, contributing valuable insights to future clinical research.

Animal models play a pivotal role in AD research. Various transgenic mouse models, such as 3xTg and 5xFAD, have been employed to investigate AD-related pathologies [32]. Recognizing the distinctions between the 5xFAD and 3xTg AD mouse models is imperative, as they capture different facets of Alzheimer’s disease pathology. In our previous study on Treg therapy for AD, we utilized 3xTg mice that express human APP695, PS1, and Tau. In these mice, amyloid deposition and memory deficits manifested at 6 and 4.5 months, respectively. Pathological changes, especially amyloid deposition, occurred at a slower pace compared to 5xFAD mice. Consequently, the 3xTg model is well-suited for investigating the later stages of AD and the interplay between amyloid and tau pathologies [33]. In contrast, 5xFAD mice rapidly develop severe amyloid pathology, with extensive Aβ accumulation primarily in the brain, commencing as early as two months of age. This model proves particularly valuable for studying the early stages of amyloid deposition and its impact on neuronal function [18]. Our study aimed to exploit these distinctions to better understand the potential of Treg therapy across different stages and aspects of AD.

Some reports on the clinical application of cell therapy suggest a correlation between the cell dose and the occurrence and severity of cytokine-release storms. In the case of CAR-T cells, the usual administration range is from 1 × 10^5^ to 1 × 10^10^ CAR-T cells/kg for patients undergoing cancer treatment [34]. In Treg therapy studies of neurodegenerative diseases such as AD, ALS, and PD, typically more than 5 × 10^5^ Tregs are transferred to mice [19,35,36]. However, in our study, we transferred only 1.5 × 10^3^, 10^4^, and 10^5^ Tregs to 5xFAD mice and observed significant neuroprotective effects even at lower doses (Figure 3 and Figure 4). Although these effects were dose-dependent, the observation that as few as 10^3^ Tregs could have therapeutic effects is promising for future clinical applications in humans.

In addition to our findings, it is crucial to highlight the methodological advancements presented in this study that significantly improved current methods for producing efficiently ex vivo-expanded Tregs. Our protocol, which employs G-rex plates for the expansion of antigen-specific Tregs, represents a substantial advancement over traditional methods. This approach not only enhances the expansion efficiency of Tregs but also ensures greater consistency and reproducibility, crucial for clinical applications. The ability to efficiently generate many antigen-specific Tregs ex vivo is a key step towards realizing the full therapeutic potential of Treg therapy in clinical settings. Moreover, the effectiveness of this method in generating Aβ-specific Tregs, as demonstrated in our study, opens new avenues for targeted immunotherapy in AD and potentially other neurodegenerative diseases. This advancement underscores our commitment to improving Treg-based therapies and sets a new benchmark for future research in this field.

## 4. Materials and Methods

### 4.1. Animals

Female 5xFAD mice, carrying transgenes encoding APP (Swedish, Florida, and London) and PS1 (M146L and L286V), along with control mice, were sourced from the Korea Research Institute of Bioscience and Biotechnology (KRIBB). C57BL/6 mice for Treg isolation were obtained from the Jackson Laboratory (Bar Harbor, ME, USA). All animals were housed in a controlled environment with a 12-h light/dark cycle and had ad libitum access to food and water. The animal experiments strictly adhered to the guidelines for animal care and the guiding principles for experiments involving animals. The University of Kyung Hee Animal Care and Use Committee approved all experimental procedures under the protocol number KHUASP(SE)-21-255. Furthermore, all animal studies underwent thorough review and were conducted in accordance with the ARRIVE guidelines [37].

### 4.2. Regulatory T Cells Manufacturing and Adoptive Transfer

Aβ-specific Treg cells were induced by culturing CD8+ T cell-depleted splenocytes with fibrillized Aβ and bee venom phospholipase A2. The Amyloid-β (Aβ; Genscript, Piscataway, NJ, USA, catalog number (*Cat No*): RP10017) fibrillization process involved diluting 5 mM Aβ1-42 with 10 mM HCl to reach a final concentration of 100 µM. The resulting mixture was incubated at 37 °C overnight [38].

Spleens obtained from C57BL/6J mice underwent mechanical disruption using a 40 µm strainer. Following red blood cell (RBC) lysis, CD8+ cells were removed through the application of CD8a (Ly-2) MicroBeads (Miltenyi Biotec, Auburn, CA, USA, *Cat No*: 130-117-044).

For antigen presentation, CD8-cells were plated in 96-well U-bottom plates with 0.5 µM fibrillized Aβ and 0.4 µg/mL bee venom phospholipase A2 (bvPLA2; Sigma Aldrich, St. Louis, MO, USA *Cat No*: P9279) for a duration of 4 days.

Following antigen presentation, CD4^+^CD25^+^ regulatory T cells (Tregs) were isolated using a CD4^+^CD25^+^ Regulatory T Cell Isolation Kit (Miltenyi Biotec, *Cat No*: 130-091-041) and expanded for an additional 2 weeks. Purified Tregs were seeded into a G-Rex 24-well plate (Wilson Wolf Manufacturing, St. Paul, MN, USA, *Cat No*: 80192M) with CD 3/28 MACSiBead™ Particles of Treg Expansion Kit (Miltenyi Biotec, *Cat No*: 130-093-627) and 2000 U/mL rmIL-2 (R&D Systems, Minneapolis, MN, USA, *Cat No*: 402-ML-020/CF).

After Treg expansion, Ex vivo-expanded Tregs were washed with media to remove IL-2, and MACSiBead™ Particles were separated by MACSiMAG™ Separator (Miltenyi Biotec, *Cat No*: 130-092-168). 1 × 10^3^, 1 × 10^4^ and 1 × 10^5^ Tregs were adoptively transferred to 5xFAD mice via intravenous.

The purity of isolated Treg was confirmed using flow cytometry after each isolation and expansion.

### 4.3. Flow Cytometry

To assess the purity of isolated cells, FITC-CD8 (Invitrogen, Carlsbad, CA, USA, *Cat No*: 11-0081-82) staining was performed after CD8 depletion. Following Treg isolation and expansion, PE-Cy5-CD4 (Invitrogen, *Cat No*: 15-0041-82) and APC-cy7-CD25 (BD Pharmingen, Franklin Lakes, NJ, USA, *Cat No*: 557658) staining was conducted in the dark at 4℃ for 30 min. Subsequently, the samples were washed with BD FACS Stain buffer (BD Bioscience, San Jose, CA, USA, *Cat No*: 554656) and subjected to flow cytometry analysis. Data acquisition was carried out using a BD FACSlyric™ flow cytometer (BD bioscience, reference number (REF): 659 180), and analysis was performed using BD FACSuite software (BD bioscience, v1,2,1,5657).

### 4.4. Behaviour Test

Eight weeks after the transfer of Tregs, we conducted the Y-maze test to evaluate spatial memory in mice. The test, lasting 5 min, took place in a Y-shaped maze with three black opaque plastic arms positioned 120° apart. After entering the maze’s center, mice were permitted to explore the three arms. We calculated the percentage of alternations by tracking the number of arm entries and triads, defining an entry as when all four limbs were within the arm. This test serves as a metric for quantifying cognitive deficits in mice and assessing the impact of novel treatments on cognition [39].

For the assessment of aversive memory, which stimulates amygdale and hippocampus, we employed the passive avoidance test (PAT), utilizing an apparatus measuring 20 × 20 × 30 cm, comprising a lit chamber and a dark chamber separated by a door. On training days, mice were positioned in the lit chamber facing away from the door. Upon entry into the dark chamber, the door closed, and a foot shock (0.35 mA, 2 s) was administered. Thirty seconds after the shock, the mice were euthanized. This trial was conducted over two days. On the test day, mice were placed in a light chamber, and the latency to enter the dark chamber was recorded within a 5-min window [40].

### 4.5. Immunofluorescence Analysis

For immunofluorescence, mice were anesthetized with isoflurane (Ifran Solution; Hana Pharm Co., Seoul, Republic of Korea, *Cat No*: 3003) and transcardially perfused with PBS. Brains were harvested and divided into two equal parts. One half of the brain was postfixed in 4% paraformaldehyde at 4 °C overnight, then transferred to a 30% sucrose solution and subsequently frozen-sectioned into 30 μm thick coronal sections using a cryomicrotome (HM525 NX; Thermo Fisher Scientific, Inc., Waltham, MA, USA, *Cat No*: 95-664-0EC). The brain sections underwent a series of treatments, beginning with a 5-min incubation in 50% formic acid at room temperature (RT), followed by heating in 10 mM sodium citrate buffer (pH 6.0) at 60 °C after washing with PBS. Blocking was performed by incubating the sections with 5% bovine serum albumin in TBSTr for 30 min. Subsequently, the sections were incubated with primary antibodies, including mouse monoclonal 4G8 antibody (1:500; BioLegend, San Diego, CA, USA, *Cat No*: 800701), phospho-Tau (1:1000; Invitrogen, REF: MN1020), and NOS2 (1:500; Santa Cruz Biotechnology, Santa Cruz, CA, USA, *Cat No*: sc-7271), for 3 days at 4 °C. Following primary antibody incubation, the brain sections were washed with Tris-buffered saline containing 0.5% Triton X-100 (TBSTr) and incubated for 2 h at RT with Alexa 488-conjugated IgG secondary antibodies (Invitrogen, *Cat No*: A32723. Slides were mounted with DAPI mounting medium (Vector Laboratories, Burlingame, CA, USA, *Cat No*: VEC-H-1200) and examined under an LSM 800 confocal laser scanning microscope (Carl Zeiss, Oberkochen, Germany). Staining intensity was quantified by measuring the integral density of the region of interest from monochromatic images using ImageJ software (National Institutes of Health, USA, v1.52a). The percentage staining intensity was calculated relative to 5xFAD and multiplied by 100 for normalization purposes.

### 4.6. RNA Extraction and RT-PCR Assays

Total RNA was extracted from hippocampal brain tissue using the easy-BLUE RNA extraction kit (iNtRON Biotechnology, Republic of Korea, *Cat No*: 17061). Subsequently, cDNA synthesis was carried out with Cyclescript reverse transcriptase (Bioneer, Republic of Korea, *Cat No*: E-3131). Real-time RT-PCR samples were prepared using the SensiFAST SYBR no-Rox kit (Bioline, Republic of Korea, *Cat No*: BIO-98005), and the amplification was conducted on the CFX Connect System (Bio-Rad, Hercules, CA, USA, *Cat No*: 1855201).

The cycling conditions comprised an initial denaturation cycle at 95 °C for 30 s, followed by 49 cycles of denaturation at 95 °C for 10 s, annealing at 55 °C for 30 s, and a final melting curve stage at 95 °C for 10 s, 50 °C for 5 s, with a gradual increase until reaching 95 °C. The quantification of target mRNAs was normalized to the expression levels of mouse β-actin, a designated housekeeping gene serving as an endogenous control. All fold changes are expressed relative to the WT. The primer sequences are provided in Table 1.

### 4.7. Statistical Analysis

The statistical analyses were conducted utilizing GraphPad Prism 5.01 software (GraphPad Software Inc., San Diego, CA, USA). A one-way analysis of variance (ANOVA) was employed, followed by Tukey’s multiple comparison test. All experiments were conducted in a blinded manner and independently repeated under identical conditions. Statistical significance was considered at *p* < 0.05.

## Figures and Tables

**Figure 1 ijms-25-00783-f001:**
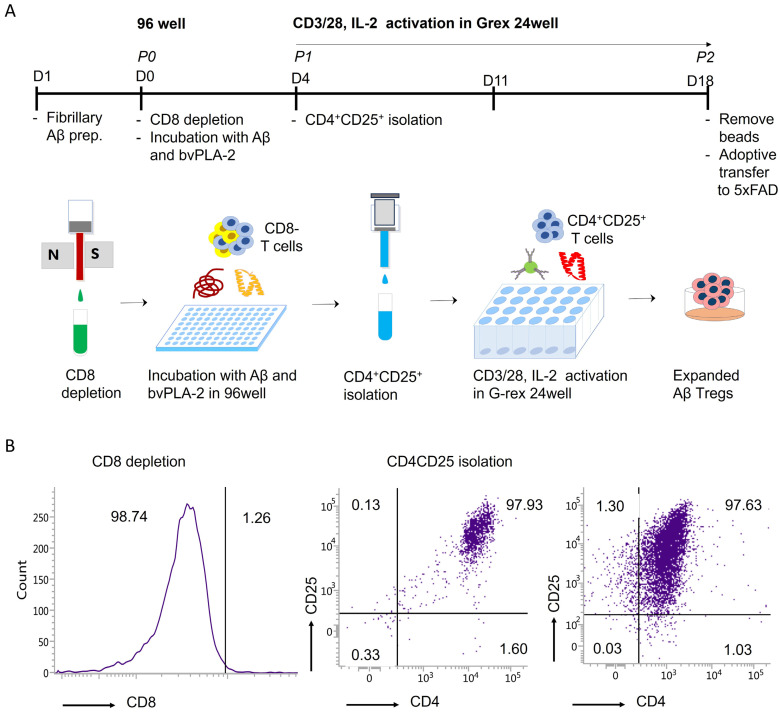
Ex vivo expansion of Aβ-specific Treg. (**A**) Schematic representation of the ex vivo expansion of Aβ-specific Treg. (**B**) The purity of isolated Aβ-specific Treg was analyzed by flow cytometry. Aβ-specific Tregs were stained with FITC-CD8 after CD8 depletion and PE-Cy5-CD4 and APC-cy7-CD25 after Treg isolation and expansion.

**Figure 2 ijms-25-00783-f002:**
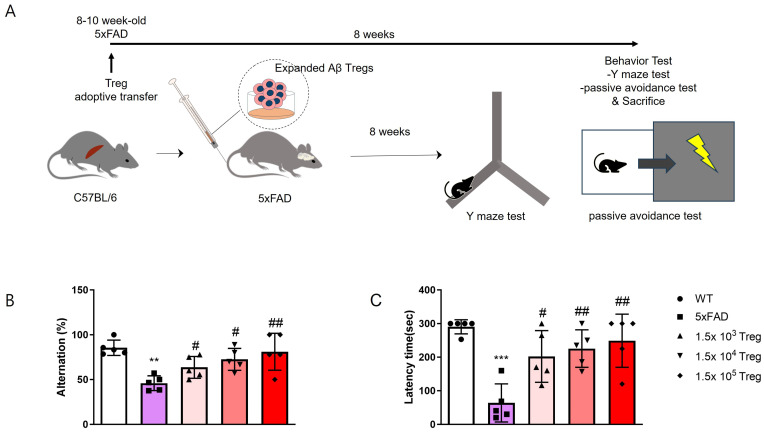
Treg transfer ameliorates cognitive dysfunction in 5xFAD mice. (**A**) Schematic representation of Treg adoptive transfer to 5xFAD mice and behavior test. (**B**) The alternation of Y maze test. (**C**) The latency time in the passive avoidance test (** *p* < 0.01, *** *p* < 0.001 vs. the WT group and # *p* < 0.05, ## *p* < 0.01, vs. the 5xFAD group, one-way ANOVA, Newman-Keuls multiple comparison test (*n* = 5). Data are presented as the mean ± SEM).

**Figure 3 ijms-25-00783-f003:**
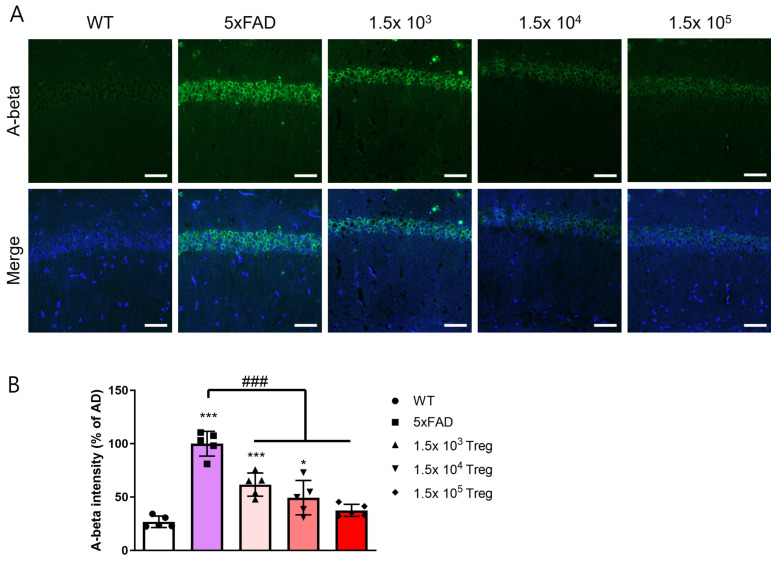
Treg transfer reduces the accumulation of Aβ in 5xFAD mice. (**A**) The expression levels of Aβ in the CA1 of the hippocampus were assessed with immunostaining. (**B**) Aβ intensity in the CA1 of the hippocampus (* *p* < 0.05, *** *p* < 0.001 vs. the WT group and ### *p* < 0.001 vs. the 5xFAD group, one-way ANOVA, Newman–Keuls multiple comparison test (*n* = 5). Data are shown as the mean ± SEM).

**Figure 4 ijms-25-00783-f004:**
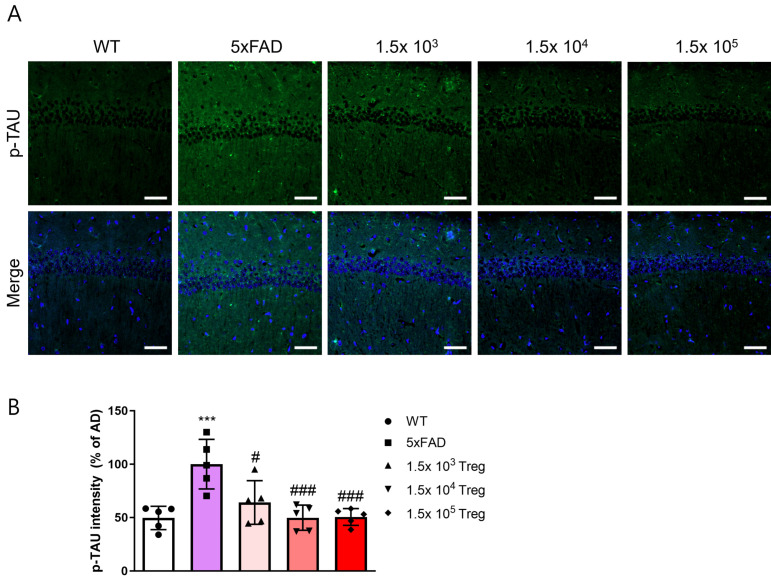
Treg transfer reduces the accumulation of p-TAU in 5xFAD mice. (**A**) The expression levels of p-TAU in the CA1 of the hippocampus were assessed with immunostaining. (**B**) p-TAU intensity in the CA1 of the hippocampus (*** *p* < 0.001 vs. the WT group and # *p* < 0.05, ### *p* < 0.001 vs. the 5xFAD group, one-way ANOVA, Newman–Keuls multiple comparison test (*n* = 5). Data are shown as the mean ± SEM).

**Figure 5 ijms-25-00783-f005:**
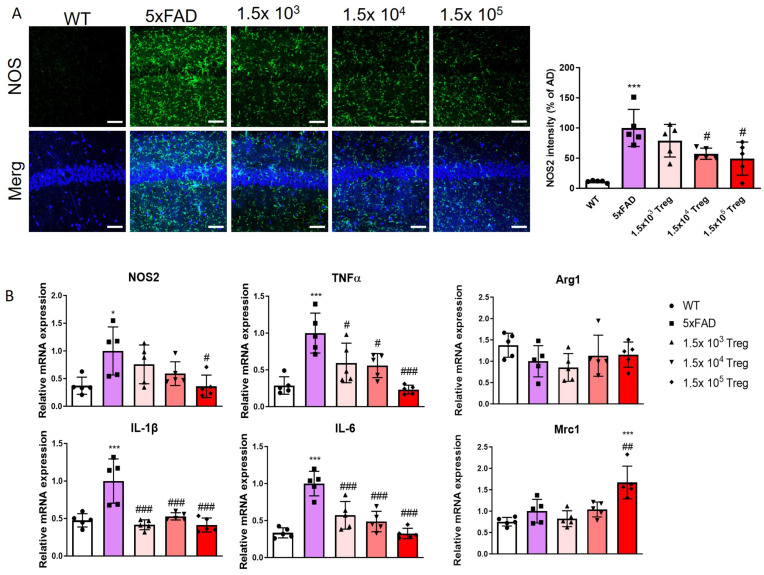
Treg transfer modulates neuroinflammation in 5xFAD mice. (**A**) The expression levels and intensity of NOS2 in the CA1 of the hippocampus assessed with immunostaining. (**B**) The mRNA expression of NOS2, TNF-α, IL-1β, Arg1, and Mrc1 (* *p* < 0.05, *** *p* < 0.001 vs. the WT group and # *p* < 0.05, ## *p* < 0.01, ### *p* < 0.001 vs. the 5xFAD group, one-way ANOVA, Newman–Keuls multiple [21] comparison test (*n* = 5). Data are shown as the mean ± SEM).

**Table 1 ijms-25-00783-t001:** Primer sequences for RT-PCR.

Gene Name	Forward Primer Sequence (5′–3′)	Reverse Primer Sequence (5′–3′)
*β-actin*	GTG CTA TGT TGC TCT AGA CTT CG	ATG CCA CAG GAT TCC ATA CC
*NOS2*	AGG ACA TCC TGC GGC AGC	GCT TTA ACC CCT CCT GTA
*TNF-α*	GGC AGG TTC TGT CCC TTT CAC	TTC TGT GCT CAT GGT GTC TTT TCT
*IL-1β*	AAG CCT CGT GCT GTC GGA CC	TGA GGC CCA AGG CCA CAG G
*IL-6*	TTC CAT CCA GTT GCC TTC TTG	GGG AGT GGT ATC CTC TGT GAA GTC
*Arg1*	CTC CAA GCC AAA GTC CTT AGA G	AGG AGC TGT CAT TAG GGA CAT C
*Mrc1*	TTC GGT GGA CTG TGG ACG AGC	ATA AGC CAC CTG CCA CTC CGG

## Data Availability

All data generated or analyzed during this study are included in this published article.

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
