# Peer review of "Therapeutic Effects of Aβ-Specific Regulatory T Cells in Alzheimer’s Disease: A Study in 5xFAD Mice"

_ijms, 2024, doi:10.3390/ijms25020783_

Round 1
Reviewer 1 Report
Comments and Suggestions for Authors
Great presentation of the topic and the objectives of the research in the introduction section.
However, description of results is very scarce. Authors should describe them with more detail what the figures display. In addition, the methods’ section lacks literature references. Authors should include some.
The authors consider Treg immunotherapy to be superior to the anti-amyloid beta antibodies aducanumab and lecanemab for the treatment of AD, due in part to safety concerns related to risks such as cerebral haemorrhage. In addition, they claim that the expansion of antigen-specific Tregs is a promising strategy that is currently under exploration in preclinical trials. In short, both are significant statements for the audience. Are they actual? should they be tempered in the wording of the article?
In any case, it is obvious that this therapy is acquiring an impressive clinical scope. Therefore, based on the content of the article and its scope, it should be considered the submission to the International Journal of Translational Medicine (another MDPI journal), in which would fit better.
Author Response
|
1. Summary |
We sincerely appreciate your time and interest in reviewing this manuscript. We will endeavor to revise and supplement the manuscript to reflect your review. Please find those revisions/corrections in the detailed response below and highlighted/ in tract changes in resubmitted files.
|
|
2. Questions for General Evaluation |
Reviewer’s Evaluation |
Response and Revisions |
|
|
Are all the cited references relevant to the research?
|
Can be improved |
Thank you for your review. We will improve it. |
|
|
Are the methods adequately described?
|
Can be improved |
Thank you for your review. We will improve it. |
|
|
Are the results clearly presented? |
Can be improved |
Thank you for your review. We will improve it. |
3. Point-by-point response to Comments and Suggestions for Authors
Comments 1: Great presentation of the topic and the objectives of the research in the introduction section.
However, description of results is very scarce. Authors should describe them with more detail what the figures display. In addition, the methods’ section lacks literature references. Authors should include some.
Response 1: Thank you for pointing this out. We agree with this comment. Following your advice, we have added a description of the results and a literature reference in the method section (Line: 108, 134,139,156,175,192)
Comments 2: The authors consider Treg immunotherapy to be superior to the anti-amyloid beta antibodies aducanumab and lecanemab for the treatment of AD, due in part to safety concerns related to risks such as cerebral haemorrhage. In addition, they claim that the expansion of antigen-specific Tregs is a promising strategy that is currently under exploration in preclinical trials. In short, both are significant statements for the audience. Are they actual? should they be tempered in the wording of the article?
Response 2: Thank you for pointing this out. Based on your adice, we adjusted the sentence structure and changed the terminology to be more euphemistic. (line: 224-244)
“Aducanumab is a monoclonal antibody designed to target amyloid beta and was approved for medical use by the U.S. Food and Drug Administration (FDA) in June 2021. Although it is the first treatment approved for Alzheimer's disease, it is controversial due to safety issues such as low clinical efficacy and side effects such as intracerebral microhemorrhage and edema [24]. Lecanemab is also a monoclonal antibody drug targeting amyloid beta and was approved by the FDA in July 2023. This is the second drug approved to reduce amyloid markers and significantly slow cognitive decline in early-stage Alzheimer's disease. However, as safety concerns have been raised about risks such as cerebral hemorrhage, long-term clinical trials are needed to confirm the efficacy and safety of lecanemab in early Alzheimer's disease [25, 26]. Overall, there is a growing need for effective AD treatments without side effects.
The efficacy of Treg immunotherapy has been demonstrated in various disease models, including graft-versus-host disease, colitis, type 1 diabetes, and multiple sclerosis [27-30]. Specifically, the expansion of antigen-specific Tregs is a potential strategy currently under exploration in preclinical trials [31]. Advancing the clinical application of Treg therapy is crucial to enhance its efficiency and establish reproducibility in patients”.
Comments 3: In any case, it is obvious that this therapy is acquiring an impressive clinical scope. Therefore, based on the content of the article and its scope, it should be considered the submission to the International Journal of Translational Medicine (another MDPI journal), in which would fit better.
Response 3:
We appreciate your assessment that this treatment has impressive clinical scope and your suggestion to recommend a journal on translational research.
However, the current paper aims to establish a protocol to produce amyloid beta-specific Tregs and is biased towards molecular biological experimental data for now.
We therefore believe it is appropriate for "this special issue", which aims to provide an updated overview of the advancements in the research on "neurodegenerative diseases: from understanding the molecular basis to developing new treatments".

Reviewer 2 Report
Comments and Suggestions for Authors
This is a well-written paper about Tregs and their therapeutics potential in an Alzheimer’s disease mouse model. However, I have some major and minor concerns about it.
Major
Your methods session isn’t well detailed. Please find some comments below:
“The Amyloid-β (Aβ; Genscript, Piscataway, NJ, USA)”, what is the reference number for it? Please, put it in your text.
“T cells (Tregs) were isolated using CD4+CD25+ 315 Regulatory T Cell Isolation Kit (Miltenyi Biotec)”. The authors must put the reference number for this kit.
Following the idea of both comments above, the authors must provide all catalog or reference numbers of the kits and reagents material used in the methods session, so your experiments can be reproducible by others.
Besides having an excellent purity of your cells (97.63), why didn’t the authors used IL-2 or other “tool” initially in the cultured media to stimulate the immune response before expanding the cells, as it is known that Treg cells will die quickly if not properly stimulated?
All the results that were presented in columns the authors must present the same bars with visible individual dots for each animal, so we can see what really happened with the groups (i.e., Fig 2B, 2C, 3B, 4B, 5A-B).
Minor
I suggest changing the sentence “Tregs can eliminate or deactivate responder cells 55 through various means, such as the secretion of 56 immunosuppressive cytokines” once the term elimination and deactivation are very strong here and not always Tregs will completely inhibit a specific mechanism whereas sometimes it can be a partial inhibition.
The sentence “Aβ-specific Treg cells were induced by culturing CD8+ T cell-101 depleted splenocytes with fibrillated Aβ and bee venom 102 phospholipase A2 [20].” Should be on your methods session and not on your results.
Y-maze evaluates mainly spatial memory and not really working memory. Please change it.
“For the assessment of learning and memory, we employed the passive avoidance test”. The Y-maze also evaluates learning and memory, and any other memory task will evaluate both. The avoidance test evaluates aversive memory, which stimulates more amygdale and hippocampus. Please, correct this information.
Authors should check for typos and grammar mistakes in the entire text.
Comments on the Quality of English LanguageAuthors should check for typos and grammar mistakes in the entire text.
Author Response
- Summary
We sincerely appreciate your time and interest in reviewing this manuscript. We will endeavor to revise and supplement the manuscript to reflect your review. Please find those revisions/corrections in the detailed response below and highlighted/in track changes in resubmitted files.
|
2. Questions for General Evaluation |
Reviewer’s Evaluation |
Response and Revisions |
|
Are the methods adequately described?
|
Must be improved
|
Thank you for your review. We will improve it. |
|
Are the results clearly presented?
|
Must be improved
|
Thank you for your review. We will improve it. |
|
3. Point-by-point response to Comments and Suggestions for Authors
|
|
Comments 1: Your methods session isn’t well detailed. Please find some comments below:
“The Amyloid-β (Aβ; Genscript, Piscataway, NJ, USA)”, what is the reference number for it? Please, put it in your text.
“T cells (Tregs) were isolated using CD4+CD25+ 315 Regulatory T Cell Isolation Kit (Miltenyi Biotec)”. The authors must put the reference number for this kit.
Following the idea of both comments above, the authors must provide all catalog or reference numbers of the kits and reagents material used in the methods session, so your experiments can be reproducible by others.
|
Response 1: Thank you for pointing this out. We agree with this comment. Therefore, we have included all catalog numbers of the kits and reagents material used in the methods session and highlighted. ( all catalog numbers and reference numbers are highlighted in line 326-436)
Comments 2:
Besides having an excellent purity of your cells (97.63), why didn’t the authors used IL-2 or other “tool” initially in the cultured media to stimulate the immune response before expanding the cells, as it is known that Treg cells will die quickly if not properly stimulated?
Response 2: Thank you for pointing this out. We instantly put the IL-2 and CD3/28 beads in the culture medium after Treg isolation and inserted them in the G-rex plate. However, that needs to be clearly ascribed in our text. So, we changed the text and highlighted it with red color and underline (line: 315-330).
“Following antigen presentation, CD4+CD25+ regulatory T cells (Tregs) were isolated using a CD4+CD25+ Regulatory T Cell Isolation Kit (Miltenyi Biotec, Cat No: 130-091-041) and expanded for an additional 2 weeks. Purified Tregs were seeded into a G-Rex 24-well plate (Wilson Wolf Manufacturing, St. Paul, MN, USA, Cat No: 80192M) with CD 3/28 MACSiBead™ Particles of Treg Expansion Kit (Miltenyi Biotec, Cat No: 130-093-627) and 2000 U/ml rmIL-2 (R&D Systems, Minneapolis, MN, USA, Cat No: 402-ML-020/CF).
After Treg expansion, Ex vivo-expanded Tregs were washed with media to remove IL-2, and MACSiBead™ Particles were separated by MACSiMAG™ Separator (Miltenyi Biotec, Cat No: 130-092-168). 1 x 103 ,1 x 104 and 1 x 105 Tregs were adoptively transferred to 5xFAD mice via intravenous.
The purity of isolated Treg was confirmed using flow cytometry after each isolation and expansion.”
Comments 3:
All the results that were presented in columns the authors must present the same bars with visible individual dots for each animal, so we can see what really happened with the groups (i.e., Fig 2B, 2C, 3B, 4B, 5A-B).
Response 3: Thanks for pointing this out. We modified the plot of the sticks with individual dots visible for each animal in the figures (Fig 2B-C, 3B, 4B, 5A-B).
Comments 4:
I suggest changing the sentence “Tregs can eliminate or deactivate responder cells through various means, such as the secretion of immunosuppressive cytokines” once the term elimination and deactivation are very strong here and not always Tregs will completely inhibit a specific mechanism whereas sometimes it can be a partial inhibition.
Response 4: Thank you for pointing this out. We agree with this comment. We modified the expression in this part to not be too strong. (line: 55-58)
“Several mechanisms of Treg-mediated suppression include the secretion of immunosuppressive cytokines by the Treg, cell-contact-dependent suppression, and functional modification or killing of antigen-presenting cell”.
Comments 5:
The sentence “Aβ-specific Treg cells were induced by culturing CD8+ T cell-101 depleted splenocytes with fibrillated Aβ and bee venom phospholipase A2 [20].” Should be on your methods session and not on your results.
Response 5: Thank you for pointing this out. Following your suggestion, we moved that sentence to the methods session. (line: 327)
Comments 6:
Y-maze evaluates mainly spatial memory and not really working memory. Please change it.
Response 6: Thank you for pointing this out. Following your suggestion, we modified that sentence (line 375)
“Eight weeks after the transfer of Tregs, we conducted the Y-maze test to evaluate spatial memory in mice”.
Comments 7:
“For the assessment of learning and memory, we employed the passive avoidance test”. The Y-maze also evaluates learning and memory, and any other memory task will evaluate both. The avoidance test evaluates aversive memory, which stimulates more amygdale and hippocampus. Please, correct this information.
Response 7: Thank you for pointing this out. Following your suggestion, we modifided that sentence (line: 384-387)
“For the assessment of aversive memory, which stimulates amygdale and hippocampus, we employed the passive avoidance test (PAT), utilizing an apparatus measuring 20 × 20 × 30 cm, comprising a lit chamber and a dark chamber separated by a door”.
|
4. Response to Comments on the Quality of English Language
|
|
Point 1: Authors should check for typos and grammar mistakes in the entire text.
Response 1: Thank you for pointing out. General grammatical mistakes in text have been improved across the board. |
